# Influence of Thermal Shocks on Residual Static Strength, Impact Strength and Elasticity of Polymer-Composite Materials Used in Firefighting Helmets

**DOI:** 10.3390/ma15010057

**Published:** 2021-12-22

**Authors:** Daniel Pieniak, Agata Walczak, Marcin Oszust, Krzysztof Przystupa, Renata Kamocka-Bronisz, Robert Piec, Grzegorz Dzień, Jarosław Selech, Dariusz Ulbrich

**Affiliations:** 1The Main School of Fire Service, Faculty of Safety Engineering and Civil Protection, Slowackiego 52/54, 01-629 Warsaw, Poland; dpieniak@sgsp.edu.pl (D.P.); awalczak@sgsp.edu.pl (A.W.); moszust@sgsp.edu.pl (M.O.); rkamocka@sgsp.edu.pl (R.K.-B.); rpiec@sgsp.edu.pl (R.P.); gdzien@sgsp.edu.pl (G.D.); 2Department of Automation, Lublin University of Technology, Nadbystrzycka 36, 20-618 Lublin, Poland; 3Department of Transport and Civil Engineering, Institute of Machines and Motor Vehicles, Poznan University of Technology, Piotrowo 3, 60-965 Poznan, Poland; jaroslaw.selech@put.poznan.pl (J.S.); dariusz.ulbrich@put.poznan.pl (D.U.)

**Keywords:** fire helmet, impact test, thermal shock, polymer composite, degradation

## Abstract

The article presents results of experimental studies on mechanical properties of the polymer-composite material used in manufacturing firefighting helmets. Conducted studies included static and impact strength tests, as well as a shock absorption test of glass fiber-reinforced polyamide 66 (PA66) samples and firefighting helmets. Samples were subject to the impact of thermal shocks before or during being placed under a mechanical load. A significant influence of thermal shocks on mechanical properties of glass fiber-reinforced PA66 was shown. The decrease in strength and elastic properties after cyclic heat shocks ranged from a few to several dozen percent. The average bending strength and modulus during the 170 degree Celsius shock dropped to several dozen percent from the room temperature strength. Under these thermal conditions, the impact strength was lost, and the lateral deflection of the helmet shells increased by approximately 300%. Moreover, while forcing a thermal shock occurring during the heat load, it was noticed that the character of a composite damage changes from the elasto-brittle type into the elasto-plastic one. It was also proved that changes in mechanical and elastic properties of the material used in a helmet shell can affect the protective abilities of a helmet.

## 1. Introduction

According to the Directive 89/656/EEC [1] that defines the main design, technological and maintenance rules of personal protective equipment, a firefighting helmet is a piece of personal protective equipment that is to protect health and provide safety at the highest possible level. Regulations concerning firefighting in Poland and other European countries allow using Type-B EN 443 helmets [2]. Such helmets are supposed to protect the head and the neck of a firefighter. The superior function of a firefighting helmet is to absorb the mechanical impact that the firefighter’s head is exposed to, redistribute absorbed kinetic energy from the direction that constitutes the greatest danger, and, in turn, to dissipate it in a way that does not cause any excessive harm to a firefighter and, if possible, does not damage a helmet itself [3]. The main role of a helmet is to protect a firefighter against mechanical risks that occur during firefighting operations and action exercises. The impact of a static mechanical load, which occurs, for instance, when a firefighter’s head is buried with rubble, is characterized by a relatively long constant value of an applied load [4]. In contrast, a dynamic load is related to a short and impulsive impact of a large amplitude [5] and is often connected with a mechanical impact applied on a helmet. Helmets commonly protect against a single impact occurring, for example, when a disconnected piece of the construction hits a helmet [6] or as a result of a direct hit in reinforcement rebars, an unnoticeable obstacle or, what rarely occurs, due to an impact of debris produced during an explosion [7]. The structure of a helmet should manage to transfer such loads, as well as absorb and dissipate mechanical energy. Mechanical loads that affect a firefighter’s head can lead to injuries within the neurocranium, close-head injuries, that constitute a greater life hazard than craniofacial injuries. The latter ones may result in serious health conditions, but they are seldom life-threatening [8]. Thus, a helmet plays a vital role as a personal protective equipment item.

Fires, especially the internal ones, are accompanied by thermal and mechanical factors. The main sources of danger for firefighters are hot surfaces and hot gases that can have a more or less intensive impact depending on heat flux density, gas or smoke composition, and temperature [9,10]. The fire environment can directly affect the personal protection of a firefighter, which is visible during extinguishing actions and action exercises. Shells of some helmets are likely to deform and melt on their surfaces [5]. Such information was confirmed by studies on firefighting helmets published in the following work [11]. The work proved a significant influence of softening a helmet shell when exposed to the hot airflow on the value of maximum forces acted on a static head model where a helmet was placed. In the case of a helmet after heat treatment, the observed values of active and passive forces redistributed to the cervical spine were lower; however, the impulse duration was longer, which is considered to be an adverse effect (this value is included in the head injury criterion (HIC) that measures the likelihood of head injury arising from an impact according to the work of [12,13]). The most external and sensitive element of a helmet is its shell [14]; thus, the following work presents studies based on an experimental assessment of the mechanical strength and stiffness of this element, which was conducted in simulated conditions. The following work defines residual strength and elasticity as the strength of a composite damaged because of thermal shocks [15]. The aim of the work is to experimentally evaluate the mechanical behavior of the helmet shell during and after heat shocks and to assess how the shell properties changed by the thermal shock translate into the entire fire helmet. The null hypothesis is that thermal shocks contribute to deteriorating properties related to strength and elasticity, both in the case of cyclic thermal shocks as well as thermal shocks related to a mechanical load.

## 2. Research Method

### 2.1. Materials

Conducted studies were based on testing samples of polyamide 66 (Ultramid A3XZG5) that is used to produce shells of firefighting helmets by injection molding. Ultramid (BASF, Ludwigshafen, Germany) is a composite material created by modifying polyamide 66 (PA66). The manufacturer labeled it with the code PA66-I GF25 FR [16]. The polyamide was modified, i.a., by adding glass fibers that constitute 25% of the overall composite mass. The modification affects the mechanical properties of the composite, improving its fire resistance and strength. Ultramid, as an intermediate product, is provided in a granulated form that is ready to use. Characteristic parameters of the composite are shown in Table 1.

Another item studied in conducted tests was a firefighting helmet and a helmet shell made of Ultramid, as shown in Figure 1. The head protection and a shell were of CALISIA VULCAN CV 102 model (KZPT Kalisz, Kalisz, Poland, currently BRANDBULL POLSKA S.A., KZPT in Kalisz, Poland).

Samples that were subject to strength tests were made by injection molding using a computer-controlled device, KM65/160/C4, equipped with a two-speed water thermostat manufactured by Wittmann. There were two kinds of samples prepared by injection to a universal mold with exchangeable mold inserts. The shape and dimensions of the first group of samples were defined based on the PN-EN ISO 527-1:2021 standard [17]. Conducted tests included samples marked “1A” in accordance with the technical norm, namely samples formed directly by injection molding with a dumbbell shape with a cross-section of 4 × 10 mm. Each subgroup included 5 tested samples (n = 5). Impact strength tests included rectangular samples with a cross-section of 4 × 23 mm (n = 5). Shells subject to the lateral stiffness tests were provided by the manufacturer. There were 4 shells tested (N = 4), one (n = 1) in every temperature range, according to Table 2.

### 2.2. Static and Impact Strength Tests

#### 2.2.1. Static Tensile Strength Testing Method

Static tensile strength tests were conducted using an electromechanical tensile testing machine Zwick/Roell Z100 equipped with a macro extensometer. The tensile test was carried out according to the ISO 527-1:2012 standard [18]. The values of specific sample parameters for particular plastics were determined based on stress-stain curves. Initial and residual (after cyclic thermal shocks) tensile strength was calculated as the maximum tensile stress value. Calculations were made using the following Equation (1):(1)$$\sigma_{M}=\frac{F_{m}}{A_{O}}$$
where:$\sigma_{M}$—tensile strength (MPa);$F_{m}$—maximum force (N);$A_{O}$—initial cross-sectional area of the sample (mm^2^).

Moreover, deformation $\varepsilon_{T}$ corresponding maximum strength (*F_m_*) was determined. Initial and residual (after thermal shocks) modulus of elasticity were calculated in terms of deformation higher than 0.05% with a tangent method. Calculations were made according to the following equation:(2)$$E_{M}=tg\alpha_{T}$$
where:$E_{M}$—Young’s modulus (MPa);$tg\alpha_{T}$—the slope of the tangent line to stress-strain diagram.

The work of stretching was calculated using the following equation:(3)$$W_{M}=\int_{l_{0}}^{l}F\left(l-l_{0}\right)$$
where:$W_{M}$—work of stretching (Nmm);F—test loading (N);$l_{0}$—initial sample length (mm);$\left(l-l_{0}\right)$—sample elongation (mm).

#### 2.2.2. Static Bending Strength Testing Method

Bending strength tests were carried out using a universal testing machine, LabTest 6.100 (Opava, Czech Republic), equipped with an integrated temperature chamber. Tests were conducted in accordance with the principles of the ISO 178 standard [18]. Bending strength was calculated using the following equation:(4)$$\sigma_{B}=\frac{3PL}{2bd^{2}}$$
where:P—test loading (N);L—distance between supports (mm); b—sample width (mm);d—sample thickness (mm).

The bending modulus was calculated using the following equation:(5)$$E_{B}=\frac{mL^{3}}{4bd^{3}}$$
where:m—slope of the diagram in a linear scope;L—distance between supports (mm);b—sample width (mm);d—sample thickness (mm).

#### 2.2.3. Shock Bending Strength Test

The shock bending strength test was conducted using impact drop towers LaborTech DFP Test 1000 (Opava, Czech Republic). The value of the potential energy of the hammer was 15 J. The test was based on registering the strength characteristic within the function of the force impulse time. The force was measured using a piezoelectric sensor. Tests were conducted on flat, notch-free samples with a cross-section of 23 × 4 mm (N = 20, n = 5). Tests were conducted under conditions of combined thermal and mechanical loads.

### 2.3. Testing Lateral Stiffness of Helmet Shells

Lateral stiffness tests were conducted in a similar way to principles included in the PN-EN 443 standard [2]. A shell was subject to quasistatic loading in width according to the direction of the AA plane defined in the technical standard. The way of mechanical loading is shown in Figure 2.

The mechanical load was imposed with a universal testing machine, LabTest 6.100 (Opava, Czech Republic), equipped with an integrated temperature chamber. The load imposed on samples was similar to the one used while testing whole helmets, but the maximum force value in tests concerning shells was lower (Figure 3); 630 N for helmets (according to the EN 443 standard) and 100 N for shells. The test was extended by introducing a thermal factor according to the parameters shown in Table 2.

### 2.4. Testing the Influence of Treating a Firefighting Helmet with a Spherical Impactor

Tests were carried out using specialized equipment of an impact hammer DPF 1000 with a static head model (Figure 4). The mechanical force was imposed in the central direction, as shown in Figure 4b. The energy of the central impact was taken to be 60 J. The impactor used in order to conduct tests was spherically ended with a diameter of 20 mm and a weight of 0.54 kg. The force was measured on the active side of a measurement track-measuring transmitter was placed on the same side as the impactor, and the force sensor was of the nominal range of 30 kN, sensitivity of 0.06 mV/N, and sampling frequency of 40 kHz.

### 2.5. Thermal Shock Loading

#### 2.5.1. Cyclic Thermal Shock Loading

When samples were tested on the static bending strength, they had been subject to cyclic thermal loads in thermal fields, being able to generate heat energy with great speed (Figure 5). That procedure was conducted at a supporting test stand that simulated actual conditions and used its module that generates the hot airflow with defined parameters. The module is equipped with an air heater HOTWIND SYSTEM (Leister Technologies AG, Sarnen, Switzerland) with heat-up power of 3680 W. This device enables to generate an airstream with the speed flow from 100 to 900 L/min and with the temperature up to 650 °C. Levels of fire heat exposure depending on time and temperature were classified. This work [7] enlists four levels of exceptional fire loads: standard, hazardous, extreme, and critical (Table 2). Based on that, as well as taking into account the need to make test cycles reflect real (operating) conditions as much as possible, the following temperature values and exposure times were defined: 20 °C (temperature in which a helmet is stored), 100 °C for 25 min, 120 °C for 15 min, 140 °C for 10 min, and 160 °C for 1 min. The highest temperature was limited to 160 °C due to the capacity of the hot airflow sensor. Temperature fields were observed and registered with a thermography camera PI400 (Optris GmbH, Berlin, Germany). Registered images are shown in Figure 5. Parameters of the hot airflow were stabilized. Figure 6 presents selected characteristics of temperature within the function of time for the nominal temperature of 140 °C. Changes in characteristics of the actual temperature (blue line) and the nominal temperature (red line) are visible. When the airflow temperature was stable, samples were placed in the operational area of the hot airflow. The ends of the samples were based on ceramic supports. Samples were placed in the operational area of the hot airflow for a particular time (Table 2).

The first series of static strength tests were conducted after cooling samples down, i.e., 48 h. Studies were extended with the assessment of residual strength and stiffness after 5 cycles of such thermal loads. Further cycles were held every 3–6 days with a simulation of firefighters’ participation in rescue and fire extinguishing actions. Tests were also conducted after cooling samples down, i.e., after 2 days.

#### 2.5.2. Thermal Shock Load in the Impact Strength, Bending and Lateral Stiffness Tests

The samples were subject to thermal processing in temperature chambers that are integrated with the impact hammer DFP 1000 and a universal testing machine LabTest 6.1000. Thermal loads were imposed according to the temperature and time values shown in Table 2. In contrast to cyclic tests, the mechanical load was imposed while curing.

#### 2.5.3. Thermal Shock Load in the Test of Treating a Firefighting Helmet with a Spherical Impactor

Thermal processing was held directly in a chamber of a drop hammer (Figure 7) that was also used in cyclic tests (Section 2.5.1). The hot airflow heating process was controlled on samples with a thermographic camera, just as in the case of cyclic tests. Based on this value, the hot airflow with the temperature of 140 °C was controlled and applied for 10 min (while heating up to the desired temperature, the airflow was directed in another direction), according to parameters shown in Table 2. Samples were treated with a spherical impactor directly after being exposed to a thermal shock.

## 3. Test Results

### 3.1. Results of Static Residual Tensile Strength Tests after Cyclic Thermal Shocks

Selected stress-strain characteristics that were determined in tensile strength tests are shown in Figure 8. Further figures present curves obtained for samples depending on the airflow temperature, exposure time, and the number of thermal shock cycles. The same range of strain and elongation axes was applied in order to facilitate comparing obtained results. Table 3 presents descriptive statistics of the results of a tensile strength test, i.e., average values, standard deviation (SD), and the coefficient of variation (CV). The results were presented with reference to the temperature of the airflow and the exposure time.

### 3.2. Results of Bending Strength Tests

Table 4 presents descriptive statistics concerning results of a bending strength test, i.e., average values, standard deviation (SD), and the coefficient of variation (CV). The results were presented with reference to the time of heating samples in a temperature chamber at the particular temperature value.

### 3.3. Results of Impact Strength Tests

Figure 9 presents a framing chart with results of impact strength tests as the maximum force-temperature (or exposure time, according to Table 2) relation. Medians of the impact strength slightly differ depending on the heat load (from the standard load to the extreme one). In this range, the impact strength is slightly decreasing. However, in the critical value (170 °C) range, the material melts up to such a point that it is not possible to measure the impact strength. At this temperature value, the polymer composite almost loses its mechanical bending strength.

### 3.4. Results of Stiffness Tests of Firefighting Helmet Shells

Table 5 presents the results of lateral stiffness tests of firefighting helmet shells. The results clearly show a significant increase in the deformation of a shell as the temperature increases. The deformation at the highest temperature value (170 °C) under F_1_ force is almost five times greater than the one observed at the standard temperature value (ok. 20 °C). A similar relation appears under F_2_ force; however, in this case, a significant increase occurs in a lower temperature range (beginning from 100 °C). In addition, residual deformations were also observed after reducing strain to F_3_ force, the greatest one at the temperature value of 170 °C.

### 3.5. Helmet Impact Test Results

Figure 10 presents force characteristics in the function of an impactor movement (the depth of penetration of an impactor into a helmet) after coming into contact with a helmet shell. Two curves in the graph represent the results of tests conducted at the standard and extreme temperature values (Table 2). Figure 11 presents deformations of firefighting helmet shells in the central area at the impact place. Similarly, as in the case of curves in Figure 10, deformations are significantly different. Tests at the standard temperature value resulted in fracturing the shell surface, which resulted in the change of the propagation direction. The main fracture divided into two directions. In the case of a shell tested at the extreme temperature value, there was a clearly visible deformation point (at the impact place); however, the surface did not fracture.

## 4. Discussion

### 4.1. Static Tensile Strength Tests (Cyclic Thermal Shocks)

Deterioration of strength parameters after a single treatment with the hot airflow is insignificant. Ultramid keeps its strength properties to a large extent. Visible changes are related to the decrease in stiffness defined by Young’s modulus. Reference [19] proved the decrease in the elastic modulus of polymer-composite materials with a polyethylene matrix after thermal shock loads. Such a state of affairs may indirectly indicate the partial recrystallization of the composite or other phase transitions that depend on the temperature value [20]. The crystallization process leads to nonequilibrium compounds that are partially crystalline. A crystalline polymer is not fully solid because it contains some amount of a liquid amorphous polymer. The consequence of creating partially crystalline compounds, where particular crystallites are of different sizes and perfection degrees, is a complex structure of such polymers [21]. The thickness of the heat-affected zone is obviously varied in the selected test conditions of thermal shocks (Table 2). The work [22] states that the heat-affected zone can consist of recrystallized or deformed spherulites and has a structure of layers with a thickness of up to approximately 100 μm. The size and distribution of these layers within the heat-affected zone may vary depending on a material type and processing conditions [20], mainly heat treatment. Slight changes in properties of the composite can indicate a limited thickness of the heat-affected zone, and changes in mechanical properties concern mainly the surface of the material. The confirmation of this can be seen in the results of studies published in the work of [23]. The work confirmed that the Vickers microhardness of the PA 66 material significantly increases after the exposure to the temperature of 140 °C and cooling down and increases even more after the exposure to the temperature of 160 °C and cooling down. The authors of the work [24] explain that the increase in crystallinity results from thermo-oxidation and refer to the earlier work [25] that proved a linear relation between microhardness and crystallinity for PP and PE. Moreover, the work [26] states that crystals are stiffer and harder than amorphous parts; thus, crystallinity and hardness change in a similar way due to thermal aging. Possible changes in properties are greater after exposure to the hot airflow. After terminating the exposure and cooling down, properties partially restored (such reasoning is also possible based on the results of static bending strength and impact strength tests). The elastic modulus is growing, and a change in the tensile strength is rather insignificant. Kagan and Roth [27] proved that for a polymer material, which is subject to thermal shock processing in a welding process with well-chosen processing conditions, the weld ratio, defined as the tensile stress at break of the welded material divided by the tensile stress at break of the bulk one, can reach 0.97 for such pristine polymers. However, they also proved that this weld ratio decreases significantly with the addition of short glass fibres in the polymer matrix, down to ~0.5 for 30% glass fiber-reinforced polymers [27], which did not occur during studies described in the following work. This confirms the probable low thickness of the heat-affected zone. Moreover, it was proved that the reason for the relatively low strength of the weld is the change in fiber orientation after welding and the low number of fibers remaining in the joining direction [28]. Observations of structures studied in the following work show that in the case of firefighting helmet shells, the mechanism of structural changes is rather different, presumably superficial. The orientation of fibers is multidirectional (Figure 12). After five cycles of thermal shocks, properties significantly deteriorate. In real conditions, during many years of maintenance, a firefighting helmet is exposed to cyclic impacts in firefighting operations. It can be assumed that changes in properties cumulate what results in uncertain protective properties is predicted but not strictly defined time of firefighting helmet maintenance. The results of static strength tests of Ultramid material exposed to cyclic thermal loads show possible damage related to fatigue and aging. Gradual damages of multicomponent compounds appear as various types of discontinuities or changes in composite properties [29]. For instance, the work [30] confirms that after conducting the aging process of PA 66 material at the temperature of 140 °C, the surface melting temperature is significantly lower. The level of periodic damage of polymer-composite materials can be expressed using phenomenological measures based on strength or stiffness damage [31]. In the discussed case, the most significant change is noticed in the elastic modulus, which directly contributes to the change in the stiffness strength. It is worth mentioning that relations, which could allow assessing the composite strength based on elastic modulus, e.g., Young’s modulus, have not been developed yet. As it is explained by Bezłowski [32], values of the elasticity moduli represent the average condition of the structure of damaged material, and the impact strength depends on damages that cause the greatest local weakening.

### 4.2. Static Tensile Strength in Operating Temperature Value

Properties concerning elasticity and strength under bending load changed due to thermal shocks. The decrease in strength and stiffness under bending load is probably connected with being plasticized due to elevated temperatures. The damage of mechanical properties with increasing temperature is related to the phase transition of the polymer chains from the glassy phase to the rubbery phase. While the flexibility of the polymer chains increases and the non-bonded interaction between the polymer chains is weakened with increasing temperature; the structure could be easily deformed under external loading [33,34]. Plastic deformation contains shear bands and crazes [35]. Moreover, the plasticization of the matrix due to a higher loading level may lead to debonding the polymer and reinforcement fibres [36]. Such a failure results in transferring the load by the matrix and damage of strength and stiffness. Thus, an important issue in engineering is to know the residual strength and stiffness of a selected composite. The damage of tensile strength and elastic modulus depending on temperature and exposure time was analyzed based on the following equations:(6)$$D_{E;R}=1-\frac{E;R}{E_{0};R_{0}}$$
where:$D_{E\left(b\right);R\left(b\right)}$—the level of strength damage (*D* = 0 for reference samples tested at the temperature of 20 °C, without thermal damages);*R*—strength value obtained in a given temperature and time range of a thermal shock;*R_0_*—strength value for reference samples tested at the temperature of 20 °C;*D_E_*—the level of stiffness damage (*D* = 0 for reference samples tested at the temperature of 20 °C, without thermal damages);*E*—elastic modulus obtained in a given temperature and time range of a thermal shock;*E*_0_—elastic modulus for reference samples tested at the temperature of 20 °C.

The speed of the damage of elastic modulus and strength was determined using the following equation:(7)$$V_{R\left(b\right)};\;V_{E\left(b\right)}=\frac{d\left[R_{\left(b\right)}/R_{0\left(b\right)}\right]}{dT};\;\frac{d\left[E_{\left(b\right)}/E_{0\left(b\right)}\right]}{dT}$$
where:*V_R_*, *V_E_*—speed of damage of strength/elastic modulus depending on the temperature increase *(V_R_* and *V_E_ = 0* for reference samples tested at the temperature of 20 °C, without thermal damages);*R—*strength value obtained in a given temperature and time range of a thermal shock;*R_0_—*strength value for reference samples tested at the temperature of 20 °C;*E—*elastic modulus obtained in a given temperature and time range of a thermal shock;*E_0_—*elastic modulus for reference samples tested at the temperature of 20 °C;*T—*temperature (°C).

Figure 13 presents graphs illustrating the level of damage of tensile strength *D*(*R_b_*) and elastic bending modulus *D*(*E_b_*). According to the damage equation, the level of losing properties can be described by a value from 0 to 1. The approximation curves of empirical results were presented. Analyses show that a proper model describing the damage that occurs in selected conditions is a logarithmic model. The value of the coefficient of determination indicates decent matching. Higher damage intensity is characteristic for elastic modulus that determines the stiffness of a helmet shell, which was additionally studied while testing lateral stiffness of whole helmets. The elastic modulus damage is relatively serious (0.66) for samples exposed to the heat load for a minute at the temperature of 170 °C. It can be concluded that the sensitivity of the composite in such thermal conditions is high or maybe even too high.

Figure 14 presents the course of changes in the damage speed using matching polynomials. The speed of actual damage is varied. The greatest increase can be seen in the first range of the temperature increase (100 °C) when the exposure time is the longest. The speed of damage is decreasing in further phases, but in the case of bending strength, it increases for temperature values higher than 140 °C. It is possible that calculated levels and speed values of the damage can be a base to assess the extent of thermal damage and decide about the suitability for use depending on the level of thermal shock hazard.

### 4.3. Impact Strength in Operating Temperature Values

Shapes of curves obtained in an impact strength test vary depending on the temperature value. Types of damage that occurred in the shock impact strength tests are described, i.a., the work of [37]. In conducted studies, two types of damage occurred. Curves developed based on results of tests at the temperature of 20 °C indicate elasto-brittle damage, and those illustrating tests at simulated operating temperatures indicate elasto-plastic damage. Softening the Ultramid material at the temperature of 170 °C made it impossible to test the impact strength. Deteriorating dynamic stiffness at elevated temperatures may be reflected in the capacity to dissipate the impact energy and accumulate the load in a smaller area, which, in turn, can increase the risk of a helmet shell perforation (Figure 15). While testing the composite at the temperature of 140 °C, the amount of expended deformation energy, up to the maximum force value, is similar to the amount expended in the further damage process. It seems that the applied way to reinforce the composite does not constitute a significant obstacle in the further process of damaging the composite [38,39]. Such a process affects the dissipation of the impact energy. Values of the average work of dynamic bending are lower in the case of samples subject to the heat load and are as follows: 5.4 J–20 °C, 4.7 J–100 °C, 3.6 J–140 °C. It can indirectly indicate a lower capacity of the composite to absorb the impact energy in operating temperatures, which has a possible impact on the shock absorption properties of the whole helmet.

### 4.4. Lateral Stiffness of a Helmet Shell

The lateral deformation of firefighting helmet shells in terms of thermal load hazards was measured (Table 2). Conducted studies in the quasistatic loading conditions showed the relation between increasing temperature values and the deformation of shells. Observed deformation was a few times greater in the case of shells exposed to the temperature of 170 °C in comparison to shells tested at the temperature of 20 °C. However, it needs to be stressed that at the temperature of 100 °C, the deformation was twice as big as that one for the temperature of 20 °C. A visual evaluation revealed a plastic deformation of a helmet. Permanent deformations are also visible after unloading helmets, which needs to be considered as an adverse effect.

### 4.5. Helmet Impact Test Results

The trajectory of force during the test (Figure 10) at the temperature of 20 °C is cyclically enhanced, which is probably related to combined displacements, deformations, and damaging materials of a shell and a shock-absorbing liner (Figure 15). The pink curve (Figure 10) illustrates results obtained after a helmet was exposed to the temperature of 140 °C for 10 min. Its trajectory is different from the one for the previous curve, and cyclic phases of enhancement are less apparent. It is probably connected with lower stiffness of a shell and better absorption of the impact energy by a deformation of a shell itself. In the elevated temperature, maximum force values are lower and amount approximately 3500 N, so they do not exceed the allowed values specified in the work of [40]. Progressive deformations in standard thermal conditions (according to Table 2) are visible. They probably appear due to the fact that because of high impact speed, the initial yield strength changes. In extreme thermal conditions (140 °C), this effect is partially reduced by increasing the vulnerability of the composite a shell is made of, i.e., softening the composite [41]. High plasticity of the composite under the influence of heat can result in serious shell deformations. As stated before, shells of firefighting helmets, as well as other protective helmets, as an external element, take an impact and absorb some part of its energy [41]. A shell should affect the force vector directed toward the head and transfer the force to different directions that are less dangerous for the especially vulnerable head of a firefighter. Great deformation can cause a higher accumulation of point load (Figure 15a). A helmet shell should also protect against the penetration of dangerous elements. Such a capacity can be impaired due to changes in properties of a helmet shell, i.a., resilience and impact strength. Serious deformations and changes in the resilience and elasticity of a shell, even without its puncture, can also contribute to damaging a liner that absorbs the shock impact. Figure 15b presents a damaged absorbing liner placed under a shell that was exposed to a temperature of 140 °C. This, in real maintenance conditions, may contribute to a greater hazard for a firefighter because of the increased load an absorber is subject to. The role of a liner (absorber) is to absorb external dynamic loads and dissipate the impact energy. This process is based on converting kinetic energy to another type of energy in a stable and controlled way. This function is similar to the function of a helmet shell itself, but the material a liner is made of absorbs a bigger part of the energy of a mechanical impact than a shell, mainly thanks to slow deformation that lasts relatively long [42]. Such a mechanism facilitates higher deformation work and effective dissipation of the impact energy. However, dissipation occurs in synergy with a helmet shell. Because an amortizing liner is located directly under a shell, these two elements stick to each other, and a liner has a shape adjusted to a shell shape [43]. In addition to its primary function, an absorber should be able to deform up to the point that provides “survival space” to a firefighter. Moreover, it should reduce accelerations [44] and redistribution of forces in the direction of the firefighter’s body [45,46]. Nevertheless, too serious deformation of a shell can cause too high a strain in the material an absorber is made of. Such strain can exceed the strength limit (Figure 15b).

## 5. Summary

Based on a specialized literature study and an analysis of the test results, the following conclusions were made:The studied helmet model is commonly used by fire brigade units in Poland. Moreover, it is exported to many countries. Considering the above and the fact that there have been few studies concerning damages that occur during the maintenance of firefighting helmets, the test results presented in the article are of great significance;Thermal shocks in the extent compatible with thermal hazards that occur during fire and rescue operations (elevated temperature and/or cyclic thermal loads) cause significant changes in strength and elastic properties of the composite that the studied helmet is made of. After five thermal cycles with the highest shock temperature, the remaining average strength decreased by 13.36% and the modulus of elasticity by 40.92%. The average bending strength during the 170 degree Celsius shock dropped 56.41% from the room temperature strength and the modulus of elasticity by 66.03%, respectively. Under these thermal conditions, the impact strength was lost, and the lateral deflection of the helmet shells increased by 306.56%;It was confirmed that changes in mechanical and elastic properties of the shell composite can have an impact on the protective properties of the whole helmet;In the light of conducted experimental studies, the authors of the work propose systematizing damages caused by thermal factors. Based on classifications that can be found in specialized literature [47,48,49,50,51] and the opinion of authors of the following work, they may be divided into two groups: (a) occasional damages that occur in elevated temperatures that are close to or exceed characteristic temperatures of the material a shell is made of (e.g., softening temperature) that cause unacceptable deterioration of properties of a helmet shell, deformations and melting the material; (b) aging damages that occur as a result of regular participation in firefighting operations and that allow continuous or controlled functioning. Recurring situations in which a helmet is exposed to non-critical temperature cause systematic loss of properties;Using computer simulation tools (e.g., S-Dyna, PamCrash, HUMOS) that allow analyzing the behavior of helmets in simulations, e.g., using virtual anthropometric test dummies [8], requires having proper knowledge about helmet parameters. Thus, determining the properties of firefighting helmet structures under operational conditions in the hot environment can be useful while modeling structures of helmets and during analyses such as the finite element method (FEM);Studies are limited by the range of studied objects (only one commonly used model of a firefighting helmet was tested), as well as the availability of materials used by manufacturers of helmets. In further work, the authors intend to study other popular models;In the light of obtained results, the null hypothesis should be considered proven.

## Figures and Tables

**Figure 1 materials-15-00057-f001:**
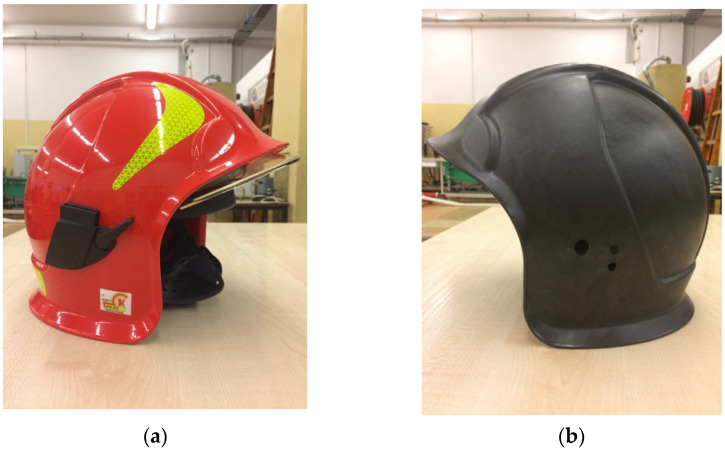
Firefighting helmet (**a**) and helmet shell (**b**,**c**) made of Ultramid (PA66-GF25FR) by injection molding.

**Figure 2 materials-15-00057-f002:**
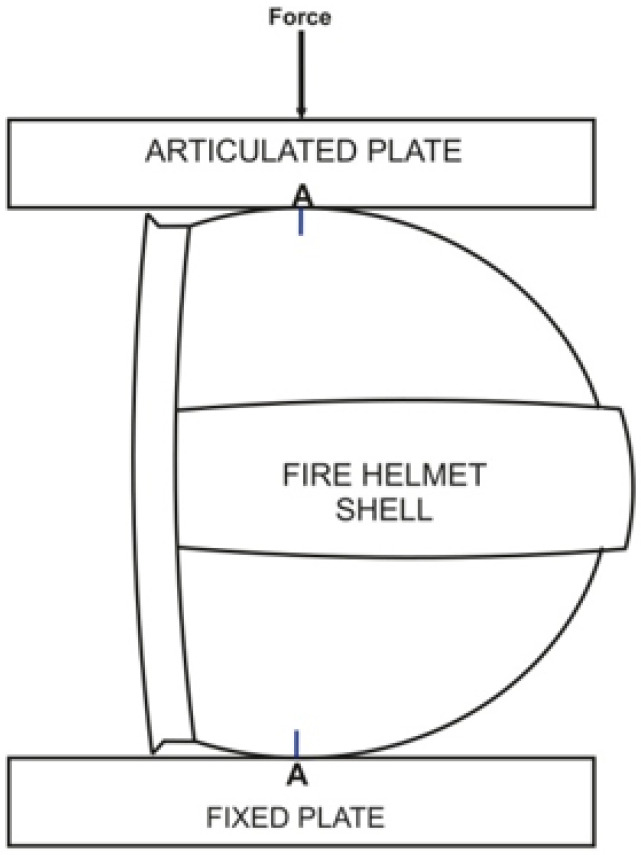
Scheme of testing lateral stiffness of helmet shells.

**Figure 3 materials-15-00057-f003:**
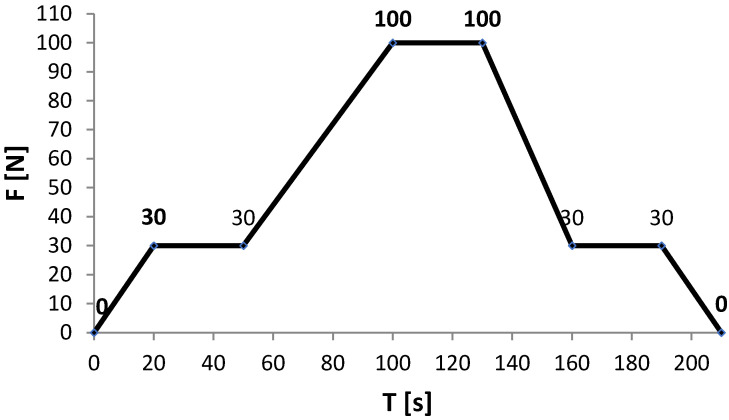
The course of mechanical load while testing lateral stiffness of firefighting helmet shells.

**Figure 4 materials-15-00057-f004:**
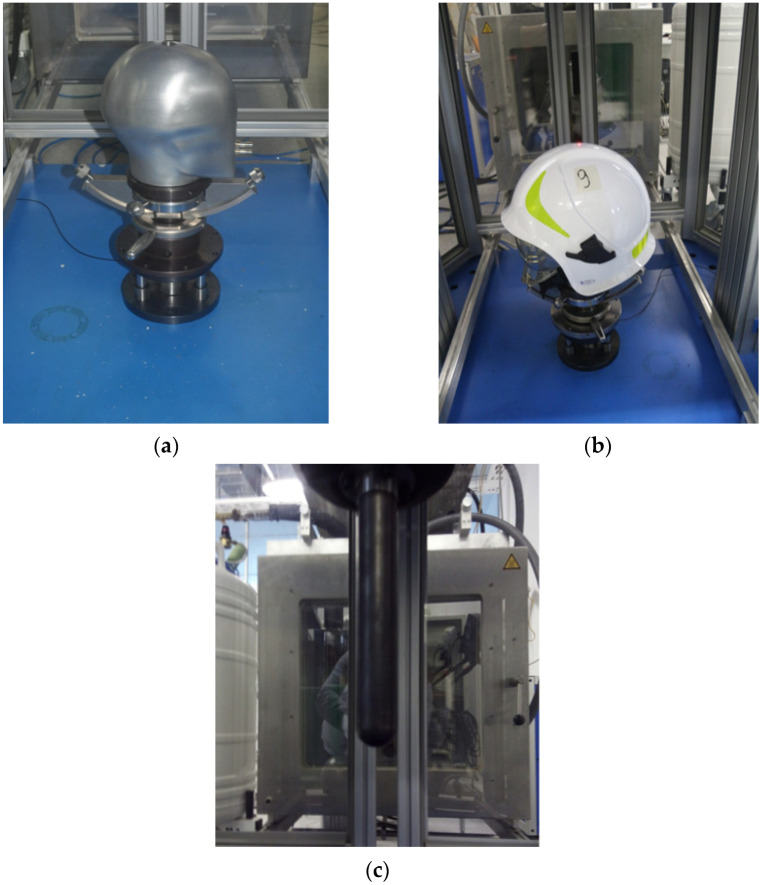
Instrumentation of an impact hammer DFP 1000 intended to test firefighting helmets: (**a**) a static head model, (**b**) the way of placing a helmet on a head model, a laser pointer indicates the point of treating a helmet with the impactor, (**c**) spherically ended impactor that hits a helmet.

**Figure 5 materials-15-00057-f005:**
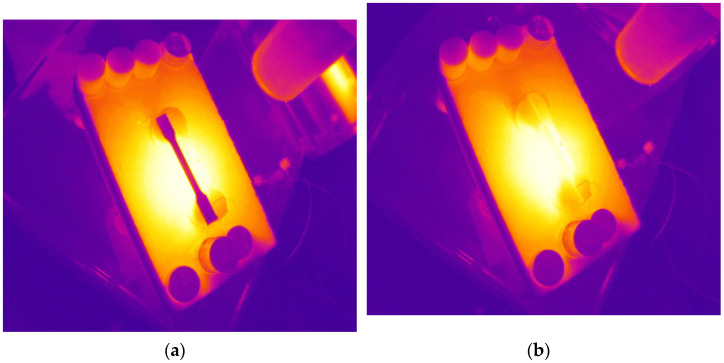
Temperature fields of a tested sample at the beginning (**a**) and at the end (**b**) of the thermal processing procedure.

**Figure 6 materials-15-00057-f006:**
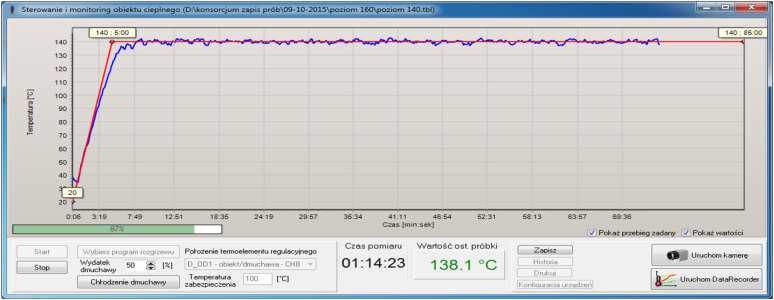
Temperature-time characteristics obtained while stabilizing parameters of a hot air flow.

**Figure 7 materials-15-00057-f007:**
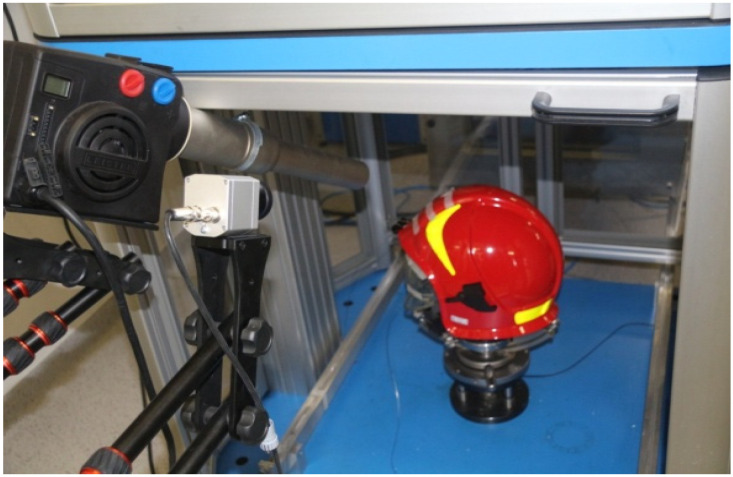
Firefighting helmet CV102 (a helmet shell and some other elements are made of Ultramid PA66-GF25FR) placed on a static head model, located in a measurement track of a drop hammer DFP 1000, which was treated with the hot airflow (air heater HOTWIND SYSTEM).

**Figure 8 materials-15-00057-f008:**
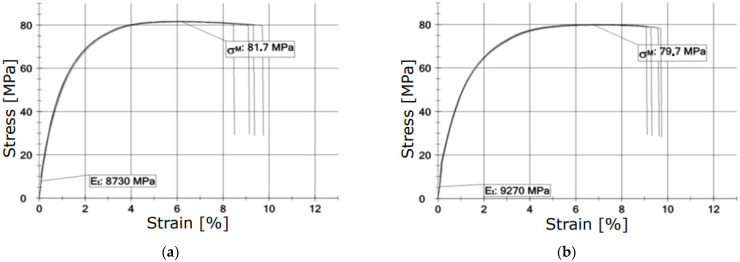
Selected characteristics σ−ε obtained in a tensile strength test after the exposure to thermal shocks: (**a**) 20 °C, (**b**) 160 °C—1 min, (**c**) 5 cycles × 100 °C—25 min, (**d**) 5 cycles × 160 °C—1 min.

**Figure 9 materials-15-00057-f009:**
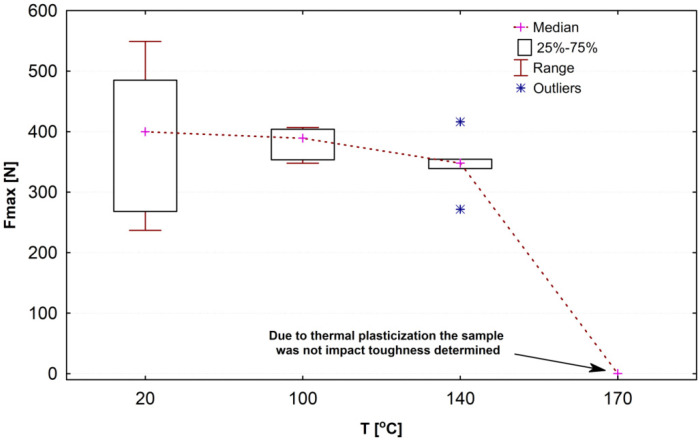
Framing chart presenting results of a bending strength test with a thermal shock.

**Figure 10 materials-15-00057-f010:**
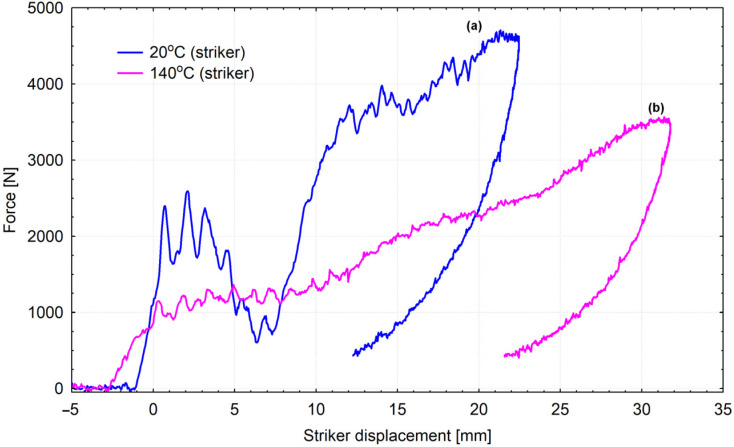
Experimental characteristics of force in the function of an impactor movement depending on thermal shock: (**a**) standard temperature value (20 °C), (**b**) extreme temperature value (thermal shock at the temperature of 140 °C for 10 min).

**Figure 11 materials-15-00057-f011:**
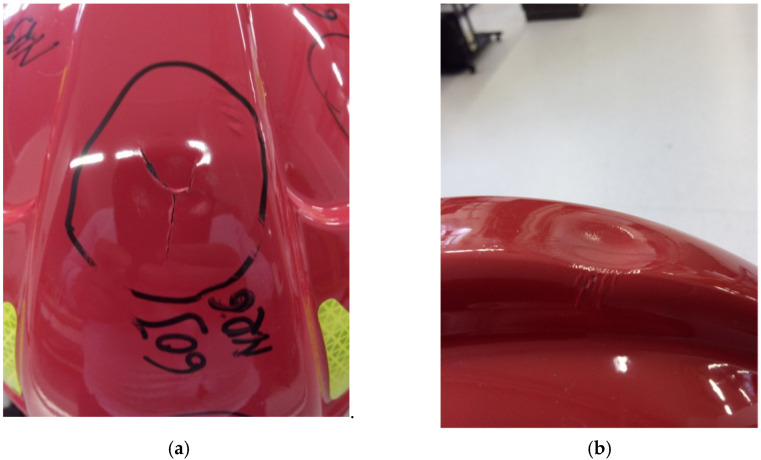
Crack and deformation of the CV102 helmet surface: (**a**) depending on thermal shock standard temperature value (20 °C), (**b**) extreme temperature value (thermal shock at the temperature of 140 °C for 10 min).

**Figure 12 materials-15-00057-f012:**
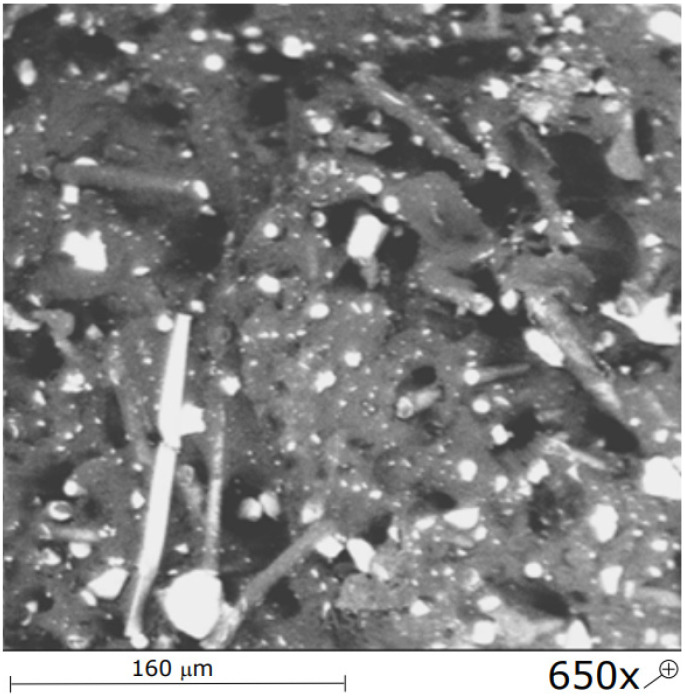
SEM image of a fracture of Ultramid sample exposed to the heat load of 100 °C.

**Figure 13 materials-15-00057-f013:**
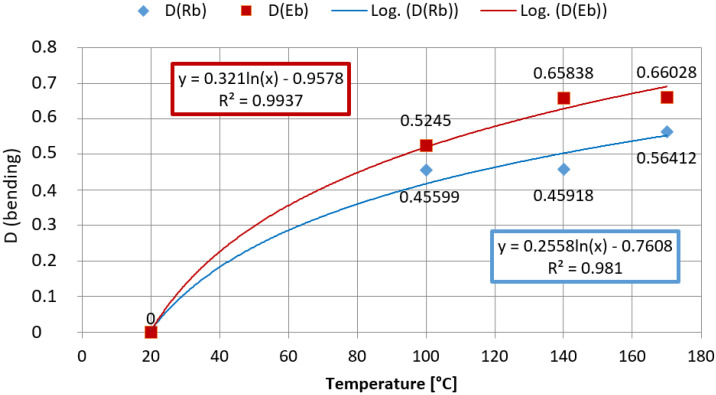
Results of testing the level of damage of strength and stiffness of the PA66 composite reinforced with glass fiber at fire temperature values: bending test.

**Figure 14 materials-15-00057-f014:**
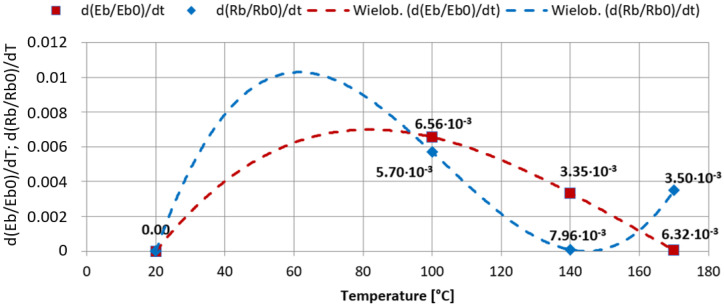
Results of testing the speed of damage of bending strength and elastic modulus depending on a temperature value.

**Figure 15 materials-15-00057-f015:**
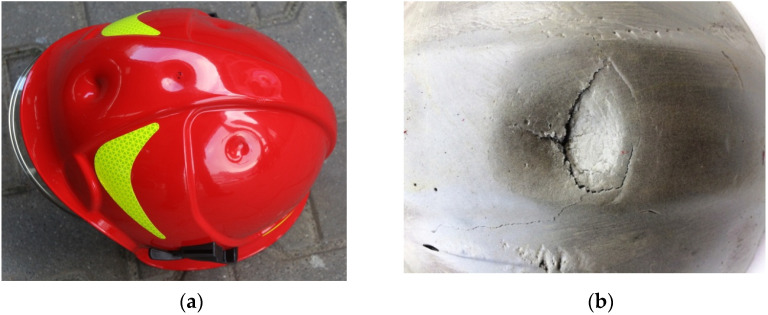
Structural elements of a firefighting helmet after the exposure to the impact loading: (**a**) a helmet shell after an impact test with the energy of 60 J using an impactor with the diameter of 20 mm at the temperature of 140 °C, (**b**) an amortizing liner at the impact point in a helmet exposed to the heat load of 140 °C.

**Table 1 materials-15-00057-t001:** Characteristic parameters of the Ultramid composite as given by the manufacturer (BASF) [16].

Ultramid PA66-GF25FR		
Parameter	Value	Unit
Glass fiber by weight	25	%
Melting temperature, DSC (ISO 3146)	260	°C
Density (ISO 1183)	1.32	g/cm^3^
Moisture absorption, equilibrium 23 °C/50% r.h. (ISO 62)	1–1.4	%
Deflection temperature 1.8 MPa (HDT A) (ISO 75-2)	240	°C
Elastic modulus in dry condition	6500	MPa

**Table 2 materials-15-00057-t002:** Technical parameters affecting a firefighter [7].

Character of Firefighter’s Work	Duration of Firefighter’s Work	Temperature
Standard	-	20 °C
Hazardous	25 min	100 °C
Extreme	10 min	140 °C
Critical	1 min	170 °C

**Table 3 materials-15-00057-t003:** Descriptive statistics of results of Ultramid tensile test after being treated with the hot airflow.

Temperature	Time	n	Mean	SD	CV
(°C)	(min)				(%)
Tensile strength $\sigma_{T}$ (MPa)					
20	-	5	81.6	0.139	0.17
100 (1 cycle)	25	5	78.8	0.666	0.85
100 (5 cycles)	25	5	70.3	0.914	1.30
120	15	5	80.0	0.754	0.94
140 (1 cycle)	10	5	80.4	0.564	0.7
140 (5 cycles)	10	5	70.5	0.348	0.49
160 (1 cycle)	1	5	79.8	0.221	0.28
160 (5 cycles)	1	5	70.7	0.418	0.59
Tensile modulus *E_T_* (MPa)					
20	-	5	8560	485	5,67
100 (1 cycle)	25	5	9270	726	7,84
100 (5 cycles)	25	5	4810	203	4,21
120	15	5	9410	804	8,55
140 (1 cycle)	10	5	9900	871	8,80
140 (5 cycles)	10	5	5230	154	2,93
160 (1 cycle)	1	5	9430	401	4,25
160 (5 cycles)	1	5	5110	242	4,74
Work to maximum force *W_T_* (Nmm)					
20	-	5	6982.82	146.82	2.10
100 (1 cycle)	25	5	7731.17	172.10	2.23
100 (5 cycles)	25	5	7117.95	193.78	2.72
120	15	5	7794.86	120.54	1.55
140 (1 cycle)	10	5	7770.21	203.90	2.62
140 (5 cycles)	10	5	7144.93	90.07	1.26
160 (1 cycle)	1	5	7640.31	81.85	1.07
160 (5 cycles)	1	5	7175.73	202.04	2.82
Elongation $\varepsilon_{T}$ (%)					
20	-	5	6.1	0.1	1.99
100 (1 cycle)	25	5	7.0	0.2	2.33
100 (5 cycles)	25	5	7.8	0.1	1.38
120	15	5	7.0	0.1	2.06
140 (1 cycle)	10	5	6.9	0.2	2.34
140 (5 cycles)	10	5	7.7	0.1	1.32
160 (1 cycle)	1	5	6.8	0.1	0.94
160 (5 cycles)	1	5	7.8	0.1	1.87

**Table 4 materials-15-00057-t004:** Descriptive statistics of results of composite bending strength test after being treated with the hot airflow.

Temperature	Time	n	Mean	SD	CV
(°C)	(min)				(%)
Bending strength $\sigma_{B}$ (MPa)					
20	-	5	119.30	1.20	1.00
100	25	5	64.90	3.60	5.50
140	10	5	64.52	10.95	16.90
170	1	5	52.00	1.50	2.90
Bending modulus *E_B_* (MPa)					
20	-	5	4481.17	431.86	9.64
100	25	5	2130.80	165.24	7.75
140	10	5	1530.84	35.13	2.29
170	1	5	1522.34	111.03	7.29
Deflection at the maximum strain $\left(\sigma_{B}\right)$ $\varepsilon_{T}$ (%)					
20	-	5	7.19	0.75	10.43
100	25	5	5.97	0.16	2.74
140	10	5	9.79	4.74	48.44
170	1	5	5.93	0.21	3.47
Deflection at the failing strain $(\sigma_{\mathrm{Broken}})\;\sigma_{\mathrm{Broken}}$ (%)					
20	-	5	10.54	0.37	3.54
100	25	5	- ^1^	-	-
140	10	5	13.57	6.61	48.66
170	1	5	-	-	-

^1^ There was no sample failure, and the deflection angle defined in the ISO 178 [19] standard after the strength decrease of 50% was not achieved.

**Table 5 materials-15-00057-t005:** Deformation of helmet shells under the influence of a mechanical load at different temperature values.

Temperature	n	Mean	Min	Max	SD
(°C)		(mm)			
Deformation of a loaded shell under F_1_ force = 30 N for 30 s					
**20 °C**	1	1.15	1.10	1.20	0.04
**100 °C**	1	2.48	1.80	3.00	0.46
**140 °C**	1	4.24	3.60	4.60	0.38
**170 °C**	1	5.36	5.00	5.60	0.26
Deformation of a loaded shell under F_2_ force = 100 N for 30 s					
**20 °C**	1	4.88	4.80	5.00	0.096
**100 °C**	1	13.04	12.40	14.20	0.713
**140 °C**	1	16.80	16.40	17.40	0.400
**170 °C**	1	19.84	19.40	20.20	0.297
Deformation of a loaded shell under F_3_ force = 30 N for 30 s					
**20 °C**	1	2.03	1.90	2.15	0.12
**100 °C**	1	7.94	7.40	9.00	0.66
**140 °C**	1	8.48	7.20	8.80	0.72
**170 °C**	1	10.52	9.40	11.20	0.69

## Data Availability

The data presented in this study are available on request from the corresponding author.

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
