# Peer review of "Influence of Thermal Shocks on Residual Static Strength, Impact Strength and Elasticity of Polymer-Composite Materials Used in Firefighting Helmets"

_materials, 2021, doi:10.3390/ma15010057_

Round 1

Reviewer 1 Report

The study deals with a description of a very specific technical problem - experimental assessment of the mechanical property degradation of firefighting helmets due to thermal shock treatments. The helmet material, its design and manufacturing route are standard and do not pretend on novelty. The motivation and the aim of the present study need to be clearly defined in the Introduction section. 

In Abstract and conclusions, add quantitative comparisons of property changes (e. g. in %) of pristine and thermally treated samples and helmets.  

The paper is well drafted, however it looks like a technical report with too many technical details and testing photos, which do not provide any original information (e.g. p. 4-6, Fig. 2,3,4, 8, 9,10 could be omitted).   

In conclusions, give provide the main findings in numbers (e.g. % change of the mechanical properties), not only descriptive text.  

Author Response

R: The study deals with a description of a very specific technical problem - experimental assessment of the mechanical property degradation of firefighting helmets due to thermal shock treatments. The helmet material, its design and manufacturing route are standard and do not pretend on novelty. The motivation and the aim of the present study need to be clearly defined in the Introduction section. 

A: Authors took into account the reviewer's suggestion and added the purpose of the work.

R: In Abstract and conclusions, add quantitative comparisons of property changes (e. g. in %) of pristine and thermally treated samples and helmets.  

A: The authors took into account the suggestions of the reviewer and added the percentages of decreases in mechanical properties.

R: The paper is well drafted, however it looks like a technical report with too many technical details and testing photos, which do not provide any original information (e.g. p. 4-6, Fig. 2,3,4, 8, 9,10 could be omitted).   

A: The suggestion of the reviewer was taken into account.

R: In conclusions, give provide the main findings in numbers (e.g. % change of the mechanical properties), not only descriptive text.  

A: In the section conclusions, the reviewer was taken into account and the main numerical findings (percentages) were added.

The authors thank the reviewer for contributing to the improvement of the article.

Reviewer 2 Report

This article reports detailed studies on mechanical properties of the glass fiber reinforced polyamide 66, which is commonly used as the shell of firefighting helmets. Mechanical measurements are carried out by static tensile and bending strength test, impact strength test, lateral stiffness test, and thermal shock loading, providing a comprehensive insight into the mechanical performance of the studied composite. Based on the quantitative analysis of stress-strain or force-displacement curves and the visual evaluation, the failure mechanisms are clearly elucidated. I feel the manuscript is within the scope of Journal, and it can be accepted for publication after revision. Some of the minor comments are listed below.

1. Line 79: “…by adding glass fiber that constitute 25%...” (grammar error). Should be “…by adding glass fiber that constitutes 25%...” 

2. Line 83: It should be “Table 1” instead of “Figure 1”.

3. Line 116 and 119: I suggest to define “Fm” as maximum force/load. “Strength” is misleading as the unit is “N” but not “N/m2”.

4. Line 205: “In the work [7] enlists four levels…” (grammar error). Should be “This work [7] enlists four levels…”

5. Line 240: The full stop is missing in “cyclic tests (Section 2.5.1) The hot airflow”. Should be “cyclic tests (Section 2.5.1). The hot airflow”

6. Line 264: “Ultramid bending tensile test” may not be correct. Should be “Ultramid tensile test”

7. Line 331: “approx..” (typo). Should be “approx.”

8. Line 388-389: “Plastic deformation of polymers is related to the appearance of shear bands and fractures…” Strictly, it is not correct to say “fracture” here, instead more accurately, plastic deformation contains shear bands and crazes.

9. The denominator in Eq. (7) should be dT instead of dT.

Author Response

Responses for reviwer 2:

This article reports detailed studies on mechanical properties of the glass fiber reinforced polyamide 66, which is commonly used as the shell of firefighting helmets. Mechanical measurements are carried out by static tensile and bending strength test, impact strength test, lateral stiffness test, and thermal shock loading, providing a comprehensive insight into the mechanical performance of the studied composite. Based on the quantitative analysis of stress-strain or force-displacement curves and the visual evaluation, the failure mechanisms are clearly elucidated. I feel the manuscript is within the scope of Journal, and it can be accepted for publication after revision. Some of the minor comments are listed below.

R: 1. Line 79: “…by adding glass fiber that constitute 25%...” (grammar error). Should be “…by adding glass fiber that constitutes 25%...”

A: The reviewer's suggestion was taken into account.

R: 2. Line 83: It should be “Table 1” instead of “Figure 1”.

A: The reviewer's suggestion was taken into account.

R: 3. Line 116 and 119: I suggest to define “Fm” as maximum force/load. “Strength” is misleading as the unit is “N” but not “N/m2”.

A: The reviewer's suggestion was taken into account.

R: 4. Line 205: “In the work [7] enlists four levels…” (grammar error). Should be “This work [7] enlists four levels…”

A: The reviewer's suggestion was taken into account.

R: 5. Line 240: The full stop is missing in “cyclic tests (Section 2.5.1) The hot airflow”. Should be “cyclic tests (Section 2.5.1). The hot airflow”

A: The reviewer's suggestion was taken into account.

R: 6. Line 264: “Ultramid bending tensile test” may not be correct. Should be “Ultramid tensile test”

A: The reviewer's suggestion was taken into account.

R: 7. Line 331: “approx..” (typo). Should be “approx.”

A: The reviewer's suggestion was taken into account.

R: 8. Line 388-389: “Plastic deformation of polymers is related to the appearance of shear bands and fractures…” Strictly, it is not correct to say “fracture” here, instead more accurately, plastic deformation contains shear bands and crazes.

A: The reviewer's suggestion was taken into account.

R: 9. The denominator in Eq. (7) should be dT instead of dT.

A: The reviewer's suggestion was taken into account.

The authors thank the reviewer for contributing to the improvement of the article.

Round 2

Reviewer 1 Report

The manuscript has been considerably improved: the aim of the study is clearly stated and comparative analysis of the main mechanical characteristics is provided. 

I would recommend this manuscript for publication in Materials journal.